# Systemic and Local Leptin Resistance in Patients with Cardiovascular Diseases

**DOI:** 10.3390/diagnostics15212772

**Published:** 2025-10-31

**Authors:** Olga Gruzdeva, Evgeniya Gorbatovskaya, Yulia Dyleva, Sofya Dolmatova, Anastasiya Romanova, Elena Fanaskova, Roman Tarasov, Aleksandr Stasev, Olga Barbarash

**Affiliations:** Research Institute for Complex Issues of Cardiovascular Diseases, Academician Barbarash Blvd., 6, 650002 Kemerovo, Russia; o_gruzdeva@mail.ru (O.G.); dyleva87@yandex.ru (Y.D.); soniad_30_09@mail.ru (S.D.); romanova_an_99@mail.ru (A.R.); fanaskova70@mail.ru (E.F.); tarars@kemcardio.ru (R.T.); stasan@kemcardio.ru (A.S.); barbol@kemcardio.ru (O.B.)

**Keywords:** leptin resistance, myocardial infarction, coronary heart disease, acquired heart disease, local fat depots of the heart

## Abstract

**Background/Objectives****:** The causes and mechanisms underlying the development of leptin resistance (LR) in patients with cardiovascular disease (CVD) remain unknown. Investigating the characteristics of adipose tissue in patients with CVD is a relevant scientific problem that may help to uncover the missing links in the pathogenesis of LR. This study aimed to evaluate systemic and local markers of LR in patients with different forms of CVD, and to determine the prevalence and tissue-specific expression patterns that contribute to LR. **Methods:** The study included 108 patients with myocardial infarction (MI), 96 patients with chronic coronary heart disease (CHD), and 96 patients with acquired heart disease (AHD). On day 1 of admission to the hospital, leptin and leptin receptor concentrations and the serum-free leptin index (FLI) were measured. Leptin resistance (LR) was defined as a leptin level of >6.45 ng/mL and FLI of >25. In chronic CHD and AHD patients, *LEP*, *LEPR1*, *LEPR2*, *LEPR2/2*, *LEPR3*, *LEPR3/2*, and *LEPR4* expression as well as leptin and soluble leptin receptor secretion were assessed in subcutaneous (SAT), epicardial (EAT), and perivascular (PVAT) adipose tissue. **Results:** MI and chronic CHD patients are characterized by elevated leptin levels and high FLI values in the blood serum, which indicates a high prevalence of LR, in contrast to AHD patients. In chronic CHD, the LR level was highest in EAT and moderate in SAT. Reduced leptin sensitivity in EAT is underlied by decreased expression of *LEPR1*, *LEPR2*, *LEPR2/2*, *LEPR3*, *LEPR3/2*, and *LEPR4*, and increased leptin production by epicardial adipocytes, which contributes to enhancement of leptin resistance at the systemic level. **Conclusions:** A high LR rate was detected in patients with MI and chronic CHD. The identified changes in EAT lead to the development of leptin resistance in chronic CHD patients.

## 1. Introduction

Today, MI and chronic CHD are major cardiovascular diseases (CVDs) and leading causes of disability and death in the working-age population worldwide [1]. Among CVDs, aortic valve (AV) stenosis ranks third in frequency after CHD and arterial hypertension (AH) [2].

MI, CHD, and AV stenosis have common histological features and major risk factors. The early stages of the pathophysiological process underlying degenerative AV disease are similar to atherosclerosis, the main cause of MI and chronic CHD [3].

Leptin may play an important role in the development of MI, chronic CHD, and AV stenosis [4,5]. Leptin is an adipokine secreted by subcutaneous and visceral adipose tissue. It regulates energy homeostasis [6] and directly and indirectly affects the functions of the cardiovascular system (CVS) [7]. The investigation of leptin metabolic homeostasis in local cardiac adipose tissue depots is of particular interest in CHD and AV stenosis patients.

Epicardial adipose tissue (EAT) is a fat depot located between the myocardium and the visceral layer of the epicardium. EAT is not separated from the myocardium and blood vessels by fascia; therefore, EAT can exert direct and indirect effects on the cardiovascular system through paracrine and vasocrine effects [8]. Perivascular adipose tissue (PVAT) is defined as the adipose tissue surrounding blood vessels, excluding those of the central nervous system. The absence of a barrier allows biologically active substances from perivascular adipocytes to directly penetrate the vascular endothelium and exert their effects. Dysfunctional PVAT secretes pro-inflammatory cytokines and adipokines, including leptin. Elevated leptin levels promote neointimal formation. Leptin overexpression in PVAT enhances neointimal formation, independent of body weight and serum leptin levels. Elevated leptin levels in PVAT stimulate smooth muscle cell proliferation, increasing the area of vascular damage [9].

In leptin resistance (LR), a condition that is defined as impairment of the homeostatic action of leptin, the positive metabolic effect of leptin is weakened, despite its elevated levels, but some of the pleiotropic effects on the cardiovascular system are preserved [10]. According to literature, hyperleptinemia is accompanied by an accelerated atherosclerotic process in acute and chronic CHD and promotes the transition from the initial stage to the calcification stage in AV stenosis [11,12]. In experiments, supraphysiological leptin concentrations modulate expression of several vascular genes associated with atherosclerosis and abnormal angiogenesis and are involved in the calcification of valvular interstitial cells [13]. LR is also accompanied by the development of insulin resistance, enhanced pro-inflammatory and prothrombotic responses, endothelial dysfunction, and increased activity of the sympathetic nervous system, contributing to the development and progression of both CHD and AV stenosis [14,15]. However, because most results have been obtained in cell and animal models, the role of LR in CVDs remains unclear. This study aimed to evaluate systemic and local markers of leptin resistance (LR) in patients with different forms of cardiovascular disease (CVD), and to determine the prevalence and tissue-specific expression patterns that contribute to LR

## 2. Materials and Methods

The study protocol complied with the standards of the local ethical committee of the Research Institute of Cardiology and Cardiovascular Diseases and the principles of the Helsinki Declaration. The study is prospective. Patient recruitment was carried out from 2021 to 2024. The study included 300 male CVD patients aged up to 75 years. CVD patients were divided into three groups. The first group included 108 patients with ST-segment elevation MI within 24 h before admission to the hospital. The second group involved 96 patients with chronic CHD and indications for coronary artery bypass grafting (according to coronary angiography). The third group included 96 patients with acquired heart defects (AHD) who were planning to undergo the surgical correction of an isolated AV defect caused by a degenerative condition or congenital anomaly. The fourth group (controls) consisted of 40 healthy volunteers.

Exclusion criteria included patient’s refusal of examination, age over 75 years, T1DM and T2DM, decompensated CHF, uncontrolled hypertension, aortic valve stenosis associated with rheumatic disease, infective endocarditis, and severe concomitant diseases affecting the quality of the study.

Examined patients with MI, chronic CHD, and AHD were comparable in age, BMI, and risk factors, such as AH, smoking, and a family history of cardiovascular disease (*p* > 0.05). In MI, single- and double-vessel coronary artery disease was most common; in CHD, multivessel coronary artery disease was most common. Preserved ejection fraction (EF) of the left ventricle (LV) was found in about 70% and 90% of MI and CHD patients, respectively, and in more than 90% of AHD patients (Table 1).

MI and chronic CHD patients received standard drug therapy: heparin, clopidogrel, aspirin, angiotensin-converting enzyme inhibitors (ACE inhibitors), β-blockers, calcium channel blockers, nitrates, and statins. AHD patients received warfarin, β-blockers, ACE inhibitors, statins, Ca antagonists (dihydropyridine), nitrates, and diuretics (Table 1).

All patients included in the study underwent height (m) and body weight (kg) measurements with calculation of the body mass index (BMI) (body weight (kg) to height (m^2^) ratio) (kg/m^2^). On admission to the hospital, patients underwent echocardiography (EchoCG) on an Acuson 128 XP ultrasonograph (Acuson, Mountain View, CA, USA) in two-dimensional scanning mode with assessment of the diastolic function of the left ventricle (LV), geometric and functional parameters of the cardiac cavities and walls, and the right to left heart ratio. The LV ejection fraction value was calculated in M-mode. Coronary angiography was performed using the Judkins technique (1967) on an Innova angiographic unit (GE Healthcare, Wauwatosa, WI, USA). Initially, a puncture of the femoral or radial artery was performed using the Seldinger technique (1952). Xenetix-350 (Guerbet, Villepinte, France) was used as a contrast agent.

Serum leptin and soluble leptin receptor concentrations were determined by an enzyme immunoassay using BioVendor (Kernersville, NC, USA) and eBioscience (Vienna, Austria) test systems on day 1 upon admission to the hospital. The free leptin index (FLI) was calculated as the ratio of the leptin concentration (ng/mL) to the soluble leptin receptor concentration (ng/mL) multiplied by 100. LR was defined as a leptin level of >6.45 ng/mL and FLI of >25 based on the control group data.

The sources of isolated adipocytes of human subcutaneous, epicardial, and perivascular adipose tissue (SAT, EAT, and PVAT, respectively) were 3 to 5 g fat biopsies obtained during coronary artery bypass grafting (CABG) or AV stenosis correction. SAT samples were collected from subcutaneous tissue of the inferior angle of the mediastinum wound, EAT samples were collected from the right atrium and right ventricle, and PVAT samples were collected from the ascending aorta area. Collected adipose tissue (AT) fragments were placed in a Hanks’ balanced salt solution (Sigma-Aldrich, St. Louis, MO, USA) containing gentamicin (50 μg/mL), streptomycin (100 mg/mL), and penicillin (100 U/L). Then, adipocytes were isolated from AT samples in a laminar flow cabinet of biosafety class II (BOV-001-AMS MZMO Aseptic Medical Systems, Miass Plant of Medical Equipment, Miass, Russia) as described previously [16]. Isolated adipocytes were placed in a test tube. The volume of the test tube contents was adjusted to 1 mL with culture medium, and adipocytes were counted in a Goryaev chamber. Adipocytes (2.0 × 10^6^) were ejected into the wells of a sterile 24-well plate (Greiner Bio-One GmbH, Kremsmünster, Austria) , and the volume of the well contents was adjusted to 1 mL with culture medium. Isolated adipocytes were cultured for 1 day. The medium was carefully collected from the bottom of the wells before and 24 h after culture to detect leptin and its receptor by an enzyme immunoassay using Bender MedSystems GmbH (Vienna, Austria) and R&D Systems (Burlington, ON, Canada) kits.

Total RNA was isolated from adipose tissue using an RNeasy^®^ Plus Universal Mini Kit (Qiagen, Hilden, Germany) according to the manufacturer’s protocol with some modifications described previously [17]. The quantity and quality of purified RNA were assessed using a NanoDrop 2000 spectrophotometer (Thermo Fisher Scientific, Waltham, MA, USA) by measuring absorbance at 280, 260, and 230 nm and calculating the 260/280 (A260/280) and 260/230 (A260/230) ratios. RNA integrity was assessed by agarose gel electrophoresis, followed by visualization using the Gel Doc™ XR+ system (Bio-Rad, Hercules, CA, USA). Extracted RNA was stored at −70 °C.

Single-stranded cDNA was synthesized using a High-Capacity cDNA Reverse Transcription Kit (Applied Biosystems, Foster City, CA, USA) on a VeritiTM 96-well thermal cycler (Applied Biosystems). Reverse transcription was performed according to the procedure suggested by the kit manufacturer. The quantity and quality of synthesized cDNA were assessed using a NanoDrop 2000 spectrophotometer (Thermo Fisher Scientific Inc., Waltham, MA, USA). Samples were stored at −20 °C.

*LEP*, *LEPR1*, *LEPR2*, *LEPR2/2*, *LEPR3*, *LEPR3/2*, and *LEPR4* expression (Table 2) was assessed by quantitative real-time polymerase chain reaction using TaqManTM gene expression assays (ADIPOQ Hs00605917_m1, LEP Hs00174877_m1, and IL6 Hs00174131_m1, Applied Biosystems, USA) on the ViiA 7 system (Applied Biosystems). Each 20-μL reaction mixture contained 10 μL of a TaqManTM Gene Expression Master Mix (Applied Biosystems), 1 μL of a TaqManTM Gene Expression Assay (Applied Biosystems), and 9 μL of a cDNA template (100 ng cDNA + nuclease-free water) and was amplified under the following thermal cycling conditions: 2 min at 50 °C, 10 min at 95 °C, and 40 cycles of 15 s at 95 °C and 1 min at 60 °C. A 20 μL reaction mixture without cDNA template was used as a negative control. A negative control and three technical replicates were prepared for each sample. The results were normalized using the HPRT1, GAPDH, and B2M reference genes. Expression of the test genes was calculated using the Pfaffl method and presented on a logarithmic (log 10) scale as the fold change relative to control samples.

**Table 2 diagnostics-15-02772-t002:** Nucleotide sequences of leptin and leptin receptor primers.

Gene	Sequence Direction	Sequence (5′ → 3′)	Number of Nucleotides
*LEP*	Forward primer	tgtccaagctgtgcccatcc	20
	Reverse primer	ggtggagcccaggaatgaagt	21
*LEPR 1* *NM_002303.6* *(leptin receptor isoform 1 precursor)*	Forward primer	ttcttggtccagcccaccatt	21
	Reverse primer	agcagggatgtagctgagacaa	22
*LEPR 2* *NM_001003680.3* *(leptin receptor isoform 2 precursor* , *contains an alternate 3′ terminal exon)*	Forward primer	ttcttggtccagcccaccat	20
	Reverse primer	tagcagggatgtagctgagaca	22
*LEPR 2.2* *NM**_001198687.2* (*leptin receptor isoform 2 precursor*, *contains alternate 5′ UTR and 3′ terminal exon*)	Forward primer	tttcttggtccagcccaccat	21
	Reverse primer	gcagggatgtagctgagacaat	22
*LEPR 3* *NM_001003679.3 * *(leptin receptor isoform 3 precursor* , *contains an alternate 3′ terminal exon)*	Forward primer	actgttgctttcggagtgagc	21
	Reverse primer	agccagcactgtatgttcca	20
*LEPR 3.2* *NM_001198689.2* *(leptin receptor isoform 3* , *precursor contains alternate 5′ UTR and 3′ terminal exon)*	Forward primer	ttcttggtccagcccaccatt	21
	Reverse primer	agcagggatgtagctgagacaa	22
*LEPR 4* *NM_001198688.1* *(leptin receptor isoform 4 precursor* , *contains alternate 5′ UTR and 3′ terminal exon)*	Forward primer	ttcttggtccagcccaccatt	21
	Reverse primer	agcagggatgtagctgagacaa	22

Statistical data processing was performed using STATISTICA 12 and SPSS 17.0 for Windows. The Kolmogorov–Smirnov test was used to assess the population distribution pattern based on sample data. Quantitative traits are presented as absolute values (*n*) and relative values (%), and quantitative data are presented as median (Me) and the 25th and 75th quartiles of Me (Q25; Q75). Intergroup comparison of dependent groups with a non-normal distribution of traits was performed using the Friedman’s test, and pairwise comparison was performed using the Wilcoxon test. Applying a Bonferroni correction in comparison of three dependent groups, the critical significance level was set at *p* ≤ 0.013. To assess differences in quantitative traits in comparison of two independent groups, the nonparametric Mann–Whitney U-test was used. Frequency analysis was performed using 2 × 2 contingency tables. Differences were considered statistically significant at *p* < 0.05.

## 3. Results

The leptin and leptin receptor concentrations and FLI values in the AHD group did not differ statistically significantly from those in the control group. In MI, the leptin concentration was 3.2- and 3.5-fold higher than that in AHD patients and the control group, respectively. In chronic CHD, the leptin concentration was 4.6- and 5.1-fold higher than that in AHD patients and the control group, respectively. The leptin receptor level in MI patients and chronic CHD patients was 1.4- and 1.6-fold lower, respectively, than that in AHD patients and the control group. FLI was 5.3- and 6.3-fold higher in MI patients and 6.6- and 7.7-fold higher in chronic CHD patients compared with that in AHD patients and the control group, respectively, (Table 3).

The prevalence of LR was 63% in MI, 57.3% in chronic CHD, and 25% in AHD. There were statistically significant differences in the rate of LR detection between MI and chronic CHD patients and AHD patients (*p* = 0.02 and *p* = 0.03); no differences were found between MI patients and chronic CHD patients (*p* = 0.82).

Investigation of *LEP* expression in adipocytes of fat depots of various localizations revealed that the highest mRNA levels of *LEP* were in EAT of chronic CHD patients and in SAT of AHD patients. In the group of chronic CHD patients, mRNA level of the *LEP* in EAT was 1.7- and 2.3-fold higher than that in SAT and PVAT, respectively. In the group of AHD patients, expression of the *LEP* in SAT was 1.7- and 3.5-fold higher than in EAT and PVAT, respectively (Figure 1).

*LEP* expression in EAT and PVAT was 2.9- and 2.7-fold higher (*p* = 0.02, *p* = 0.04) in the chronic CHD group than in the AHD group. There were no statistically significant differences in the mRNA level of the *LEP* in SAT adipocytes between chronic CHD and AHD patients (*p* = 0.81).

Assessment of the leptin content in the adipocyte cell culture supernatant demonstrated that the leptin concentration in EAT of chronic CHD patients was 1.1- and 1.2-higher that in SAT and PVAT adipocytes, respectively, (Figure 2).

In this case, the leptin content in the supernatant of EAT cell culture was 1.9-fold higher in chronic CHD patients than in AHD patients (*p* = 0.01).

In the group of patients with heart defects, the highest leptin concentration was characteristic of SAT. For example, the leptin content in subcutaneous adipocyte culture was 2.1- and 1.1-fold higher than that in EAT and PVAT, respectively. There were no statistically significant differences in the leptin concentration in the supernatant of SAT adipocyte culture between chronic CHD and AHD patients. There were no statistically significant differences in the leptin concentration in the supernatant of SAT and PVAT adipocyte culture between chronic CHD and AHD patients (*p* = 0.32, *p* = 0.68). PVAT adipocytes of chronic CHD patients were characterized by the lowest level of leptin secretion compared with adipocyte cultures from other localizations. The leptin concentration in the supernatant of PVAT from chronic CHD patients was 1.2- and 1.1-fold lower than that in EAT and SAT, respectively. In the group of AHD patients, the leptin content in the supernatant of PVAT was 1.8-fold higher than that in EAT (Figure 2).

Correlation analysis revealed a direct significant relationship between the serum leptin concentration and expression of the *LEP* (r = 0.61, *p* = 0.03) as well as leptin secretion (r = 0.62, *p* = 0.02) in SAT of AHD patients. There was a direct significant correlation between the serum leptin concentration and expression of the *LEP* (r = 0.72, *p* = 0.02) as well as leptin secretion (r = 0.63, *p* = 0.01) in EAT of chronic CHD patients.

In chronic CHD, the *LEPR1*, *LEPR2*, *LEPR2/2*, *LEPR3*, *LEPR3/2*, *LEPR4* mRNA levels were 2.1-, 2.1-, 1.2-, 2-, 1.1-, and 1.5-fold lower in EAT than in PVAT, respectively. Expression of the *LEPR1*, *LEPR2*, *LEPR2/2*, *LEPR3*, *LEPR3/2*, *LEPR4* was 4.7-, 2.6-, 2.1, 3.8, 1.7-, and 2.1-fold was lower in EAT than in PVAT, respectively (Figure 3).

The highest *LEPR* mRNA level for all six isoforms was typical of PVAT in chronic CHD patients. There were no statistically significant differences in expression of the *LEPR* for these isoforms between EAT and PVAT.

In AHD patients, the lowest *LEPR* mRNA level was characteristic of SAT (Figure 4).

For example, *LEPR1*, *LEPR2*, and *LEPR4* expression in SAT were reduced by 1.7-, 3-, and 2.1-fold, respectively, compared with that in EAT. There were no statistically significant differences in the mRNA levels of other leptin receptor isoforms between subcutaneous and epicardial adipocytes. In the AHD group, expression of the LEPR1 in SAT was 2.5-fold lower than that in PVAT (Figure 4). The *LEPR2*, *LEPR2/2*, *LEPR3*, *LEPR3/2*, *LEPR4* mRNA levels in subcutaneous adipocytes were 3.4-, 2.8-, 1.8-, 1.9-, and 2.5-fold lower, respectively, than those in perivascular adipocytes.

Comparison of the *LEPR* mRNA level for the above-mentioned isoforms between chronic CHD and AHD patients revealed that *LEPR1*, *LEPR2*, *LEPR2/2*, *LEPR3*, *LEPR3/2*, *LEPR4* expression in EAT of the chronic CHD group was 1.5-, 2.8-, 1.5-, 2.2-, 1.9-, and 2.1-fold lower, respectively, than that in the AHD group (*p* < 0.05). *LEPR1* and *LEPR2* expression in SAT in chronic CHD patients was 2.2- and 2.3-fold higher (*p* = 0.02, *p* = 0.01), respectively, than that in AHD patients. There were no statistically significant differences in the mRNA levels of other LEPR isoforms in SAT between the groups. Expression of the LEPR1 in PVAT of CHD patients was 2.1-fold higher than that in AHD patients (*p* = 0.01). There were no statistically significant differences in other *LEPR* isoforms in PVAT between chronic CHD and AHD patients (*p* > 0.05).

These results demonstrate that the soluble leptin receptor concentration in the supernatant of PVAT was highest in both study groups. The soluble leptin receptor levels in PVAT adipocyte culture were 1.5- and 1.8-fold and 1.4- and 1.3-fold higher than those in SAT and EAT adipocyte cultures from chronic CHD and AHD patients, respectively, (Figure 5).

In the chronic CHD group, the lowest leptin receptor concentration was found in the secretome of EAT adipocytes compared with that in other adipose tissue depots. For example, the receptor level in patients with coronary pathology was 1.3-fold lower in EAT than in SAT (Figure 5). In addition, the leptin receptor concentration in the supernatant of EAT adipocyte cell culture was 2.4-fold lower in patients with chronic CHD than in patients with heart defects (*p* = 0.01).

In AHD patients, the lowest leptin concentration was found in SAT adipocytes (Figure 5). The leptin receptor content in the supernatant of SAT adipocytes was 1.1-fold lower than that of EAT adipocytes. There were no statistically significant differences in the leptin concentration in subcutaneous adipocytes between the study groups (*p* > 0.05).

Analysis of FLI in chronic CHD patients demonstrated that FLI in the supernatant of cell culture from EAT was highest compared with that from SAT and PVAT (Figure 6).

In chronic CHD patients, FLI in EAT was 1.4- and 2.7-fold higher than that in SAT and PVAT, respectively, and was 3.7-higher than that in AHD patients. FLI in SAT of CHD patients was 1.7-fold higher than that of AHD patients. In chronic CHD patients, the lowest FLI was observed in PVAT adipocytes. In addition, there were no statistically significant differences in FLI for PVAT between patients with chronic CHD and patients with heart defects (Figure 6).

## 4. Discussion

Today, the problem of high CVD prevalence is given special attention. LR can directly contribute to the development and progression of CHD and AHD. However, in the literature, there are very few data about the rate of LR in CVD.

One of the main reasons is the lack of a single method for diagnosis of leptin resistance. Also, there are no rules regulating the choice of method. In the literature, LR is most commonly diagnosed based on the leptin concentration [18,19]. High leptin levels are believed to indicate leptin insensitivity and serve as an indirect sign of LR. Hyperleptinemia is suggested to result from a disruption of the link between leptin and its receptors, which leads to LR development [20]. The second method for assessing LR is FLI that evaluated the relationship between leptin and its receptor and reflects the functional activity of leptin.

In this study, both methods were used to diagnose LR in CVD patients. The LR rate was 63% in MI, 57.3% in chronic CHD, and 25% in AHD.

Despite the balancing of patients by BMI, the rate of LR detection was significantly higher in MI and CHD patients than in AHD patients. BMI is a height–weight indicator that reflects the content of total body fat. BMI cannot be used to assess the subcutaneous and visceral adipose tissue ratio and local changes in adipocytes. A possible explanation for the high LR rate in MI and CHD patients may be increased leptin production in visceral adipose tissue, in particular in local cardiac adipose tissue depots. Now, there is data on the correlation between leptin secretion in visceral adipose tissue and the serum leptin level [21]. Hyperleptinemia observed in MI and chronic CHD patients probably decreases the production and activity of the leptin receptor. A constantly high concentration of leptin stimulates a feedback mechanism [22]. The suppressor of intracellular signal transduction (SOCS3) is activated [23], and expression of protein tyrosine phosphatase (PTP1B) is elevated [24], which leads to a disruption of intracellular leptin signaling during the formation of the leptin–leptin receptor complex. A chronic increase in the leptin level also reduces expression of short leptin receptor isoforms that are necessary for leptin transportation to a functionally active receptor isoform.

AHD patients were characterized by low prevalence of LR, despite the fact that more than 50% of them had overweight or obesity of varying degree. A number of studies have demonstrated that the serum leptin level in severe degenerative AV stenosis with preserved ejection fraction is similar to that in healthy volunteers [25]. Our findings confirm this suggestion. But there are other data. For example, a study by R. Kolasa-Trela et al. (2011) showed that the serum leptin concentration in patients with aortic stenosis without concomitant atherosclerotic vascular lesions was lower than that in the control group [26]. In contrast, Yehong Liu et al. (2019) found an increased serum leptin level in patients with aortic valve calcification [12]. Conflicting results are probably associated with differences among study groups in age, gender, and severity of AV lesion.

Degenerative AV stenosis is an active process similar to the atherosclerotic process in CHD. Aortic valve calcification and CHD are accompanied by lipid infiltration, inflammation, neoangiogenesis, and endothelial dysfunction. However, the pathophysiology of degenerative AV stenosis can be influenced by factors, such as direct involvement of resident valve cells in leaflet remodeling via phenotype switching in cells as well as changes in adipokine status. Leptin in calcified human aortic valves has been shown to be abundantly expressed and promote osteoblastic differentiation of vascular smooth muscle cells and calcification of valve interstitial cells [27,28]. Chronic leptin stimulation of human valve interstitial cells increases alkaline phosphatase (ALP) activity and ALP, bone morphogenetic protein-2 (BMP-2), and runt-related transcription factor 2 (RUNX2) expression and decreases osteopontin expression. Probably, most of the leptin that is expressed and synthesized in the valves avoids entering the systemic circulation due to binding to the leptin receptor, which prevents hyperleptinemia and LR. According to the results of our study, the leptin receptor concentration in patients with non-coronary pathology was similar to that in healthy volunteers.

To elucidate the mechanisms of leptin resistance development in CHD patients, we studied expression of *LEPR* isoforms and *LEP* as well as secretion of leptin and its soluble receptor in adipose tissue. One of the currently known mechanisms of LR development is the decrease in *LEPR* expression and leptin receptor secretion. However, there are few data on expressions of *LEPR* transferring and signaling isoforms in CVD because most data have been obtained in animal and cell models (cell lines from healthy donors). There are six variants of human *LEPR* isoforms, which we investigated in this study in CHD and AV stenosis patients.

EAT demonstrates the lowest levels of *LEPR* isoform mRNAs relative to adipose tissue of other localizations in CHD patients. In addition, expression of six *LEPR* isoforms in EAT was lower in CHD patients than in AHD patients. In AV stenosis patients, the lowest *LEPR* mRNA levels were found in SAT, but only *LEPR1* and *LEPR2* isoforms were lower than those in CHD patients. Expression of six *LEPR* isoforms in perivascular adipose tissue was highest in both groups.

*LEPR* isoforms are found in most tissues, in particular in adipose tissue [29]. Because adipocytes express leptin receptors, leptin can directly influence the development and function of adipose tissue. A study by Bornstein et al. (2000) demonstrated that in vitro incubation of adipocytes with physiological leptin concentrations promoted activation of the JAK2/STAT3 pathway [30]. The JAK2/STAT3 pathway is involved in adipocyte differentiation via transcriptional regulation of CCAAT-enhancer-binding proteins β (C/EBPβ) [31]. A decrease in the long LEPR isoform in EAT of CHD patients and in SAT of AHD patients can impair the early stages of adipogenesis. Poor differentiation of adipose progenitor cells leads to excessive accumulation of triacylglycerols in mature adipocytes, which likely promotes cellular hypertrophy. Enlarged adipose cells secrete excess free fatty acids (FFAs), ROS, and pro-inflammatory cytokines. Excessive influx of FFAs from epicardial adipocytes penetrates the adventitia and facilitates lipid accumulation in atherosclerotic plaques of the coronary arteries [32]. Inflammation of EAT can cause dysfunction in adjacent tissues, leading to impaired myocardial microcirculation, increased vascular stiffness, and left atrial dilation [33]. Reduced adipocyte differentiation capacity along with SAT hypertrophy in AHD patients can cause inflammation and oxidative stress in this adipose tissue depot. However, due to its specific location, it does not directly affect the cardiovascular system. In addition, the pro-inflammatory activity and production of ROS are lower in SAT than in EAT [34].

Adipocyte hypertrophy in CHD patients may also be due to decreased nitric oxide (NO) production. Binding of leptin to its receptor in adipose tissue increases nitric oxide synthase activity through a complex mechanism involving protein kinase A (PKA) and mitogen-activated protein kinase (p42/44 MAPK) that stimulates NO production. NO inhibits glycerol synthesis, reduces the likelihood of fatty acid re-esterification, and decreases lipid accumulation in adipocytes [35]. Delayed LEPR1 expression in EAT in CHD patients reduces NO production, inducing adipocyte hypertrophy. Because leptin release depends on the size of adipocytes, the leptin level in EAT increases, and it can directly diffuse into the myocardial walls. Chronic exposure to leptin promotes cardiomyocyte hypertrophy and fibrosis, potentiating myocardial dysfunction [36,37,38].

The impaired differentiation and hypertrophy of adipocytes, which are found in decreased *LEPR1* expression, induce a defect in glucose transporter type 4 (GLUT4) delivery to the plasma membrane and cause insulin resistance (IR) in adipose tissue [39]. The second cause of IR development may be the hyperproduction of the tumor necrosis factor-α (TNF-α). For example, a study by Jing-Ning Huan et al. (2003) revealed that deficiency of the long *LEPR* isoform in the adipose tissue of TKO-OBR mice was accompanied by an increase in TNF-α [40]. TNF-α induces insulin resistance through direct negative interference with the insulin signaling pathway, IRS1 phosphorylation, and alteration of adipocyte differentiation and metabolism. This results in suppression of several important metabolic effects of insulin, such as stimulation of glucose transport into the cell, lipogenesis, and inhibition of isoproterenol-induced lipolysis. Increased energy accumulation in adipose tissue leads to enhanced FFA release into the bloodstream, which accelerates the atherosclerotic process [37].

The main function of short *LEPR* isoforms is associated with leptin internalization and degradation [41]. The decreased expression of five short LEPR isoforms in EAT of CHD patients reduces leptin degradation in this adipose tissue depot and increases leptin levels.

In addition, EAT adipocytes of CHD patients were characterized by the lowest soluble leptin receptor concentration. The soluble leptin receptor reflects decreased *LEPR* mRNA expression. The main function of the soluble leptin receptor is to bind leptin and, thus, modulate its bioavailability. A decrease in its content in epicardial adipocytes is also accompanied by elevated leptin levels in adipose tissue, which is reflected in this study.

For example, in CHD patients, the highest *LEP* mRNA expression and leptin secretion were observed in EAT compared with those in SAT and PVAT adipocytes. Leptin production in EAT of CHD patients was statistically significantly higher than that in AHD patients. These findings are consistent with literature data. Polyakova et al. (2019) found that the leptin gene mRNA level in males with CHD was higher in EAT than in SAT [42]. Tuowei Zhang et al. demonstrated increased expression of the *LEP* gene in CHD patients compared with AHD patients and atrial septal defect patients [43]. However, there are opposite data. Baker et al. observed a lower level of *LEP* mRNA expression in EAT compared with that in SAT of CHD patients [44]. The observed contradictions may be explained by the use of different methodological approaches. Baker et al. compared EAT derived from CHD patients, who underwent CABG, with SAT derived from patients without CHD. In the present study, EAT, SAT, and PVAT samples were collected from the same patient during surgery.

In patients with aortic valve stenosis, the highest leptin production was found in SAT. Recent studies of leptin production in patients without coronary artery disease have shown that *LEP* gene expression is higher in SAT than in visceral AT [45]. Visceral AT accounts for only one tenth of the total body fat, while SAT accounts for 82–97%. Because the main source of leptin is AT, SAT probably plays a major role in leptin production.

The lowest *LEP* mRNA level and decreased leptin content were found in PVAT of both CHD and AHD patients. This result is probably related to the fact that PVAT surrounding the coronary arteries is represented by small adipocytes with low differentiation and, thus, with lower *LEP* expression than in SAT and EAT [46]. The low leptin levels in perivascular adipocytes correspond to expression in this adipose tissue depot.

Study limitations. In this study, BMI was used to assess the presence and severity of obesity in patients with MI, chronic CHD, and AHD. Significant advantages of this indicator include simplicity and no additional costs or risks for patients. BMI is a standardized indicator of obesity, which is widely used in clinical practice, and its reference limits are established by the WHO. Despite the advantages of this method, there are limitations in its use. BMI evaluates body weight deviation from the established normal limits, without adjusting for the ratio of muscle, bone, and fat weight. BMI does not provide information on the localization and quantification of adipose tissue. Due to the above limitation, we plan to assess the rate of LR detection in patients with MI, chronic CHD, and AHD, depending on visceral obesity.

## 5. Conclusions

Patients with MI and chronic CHD have a high incidence of systemic LR. LR was established based on high leptin levels and FLI values in the serum of these patients. When assessing local LR in patients with chronic CHD, the highest level of LR was found in the EAT, a moderate level of LR was found in the SAT, and LR was absent in the PVAT. Reduced leptin sensitivity in EAT is associated with decreased *LEPR1*, *LEPR2*, *LEPR2/2*, *LEPR3*, *LEPR3/2*, and *LEPR4* expression and enhanced leptin production by epicardial adipocytes, which contributes to the intensification of LR at the systemic level. Patients with aortic valve stenosis had a low prevalence of systemic and local LR.

## Figures and Tables

**Figure 1 diagnostics-15-02772-f001:**
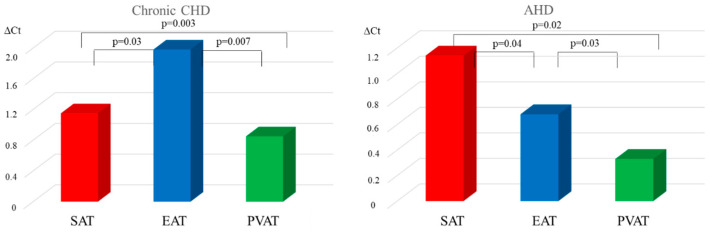
Leptin expression in local cardiac adipose tissue depots in chronic CHD and AHD patients.

**Figure 2 diagnostics-15-02772-f002:**
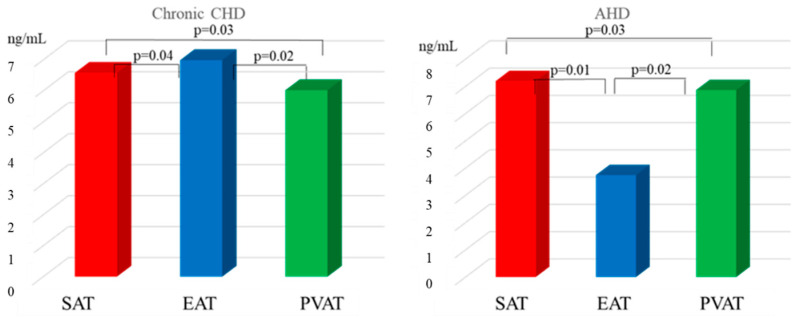
Leptin secretion in local cardiac adipose tissue depots in chronic CHD and AHD patients.

**Figure 3 diagnostics-15-02772-f003:**
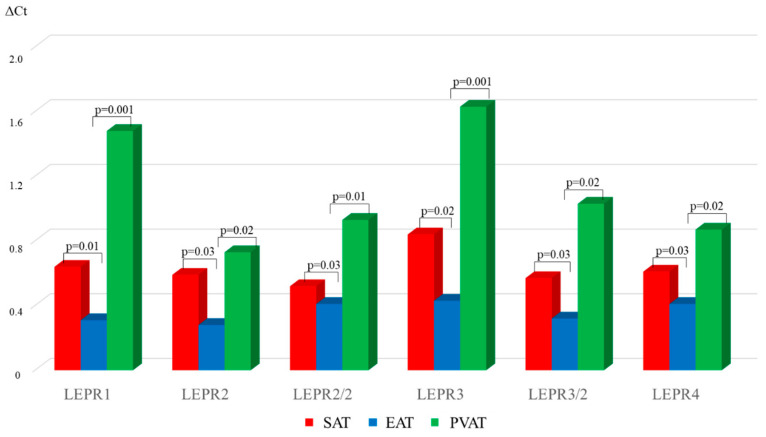
Expression of *LEPR* isoforms in local cardiac adipose tissue depots in chronic CHD patients.

**Figure 4 diagnostics-15-02772-f004:**
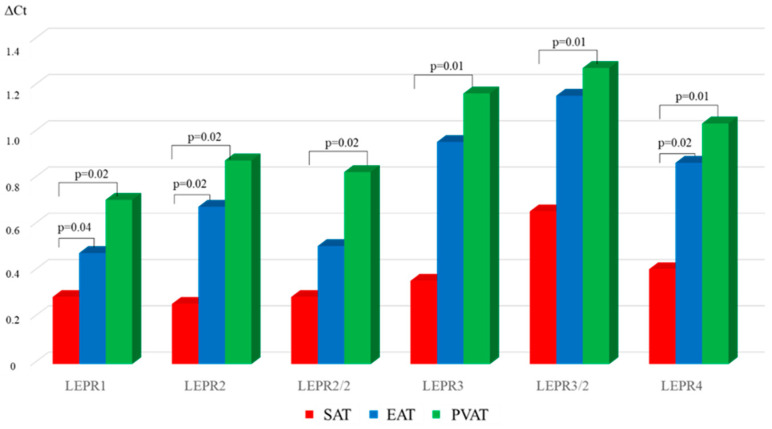
Expression of *LEPR* isoforms in local cardiac adipose tissue depots in chronic AHD patients.

**Figure 5 diagnostics-15-02772-f005:**
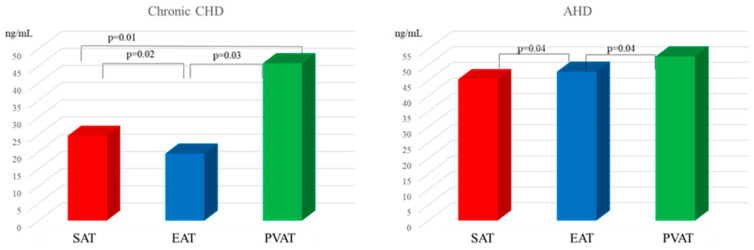
Leptin receptor secretion in local cardiac adipose tissue depots in chronic CHD and AHD patients.

**Figure 6 diagnostics-15-02772-f006:**
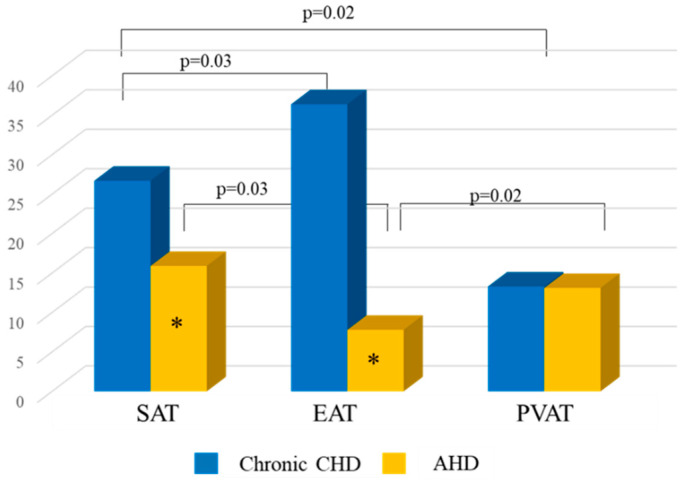
FLI of adipocytes of different localizations in chronic CHD and AHD patients. *—statistically significant differences between chronic CHD and AHD patients, *p* < 0.05.

**Table 1 diagnostics-15-02772-t001:** Clinical and anamnestic characteristics of examined patients.

Parameter	MI *n* = 108	Chronic CHD *n* = 96	AHD *n* = 96	Control Group *n* = 40
Male gender, *n* (%)	108 (100)	96 (100)	96 (100)	40 (100)
Age, years	61	64	65	58
	(57.0; 71.0)	(58.0; 69.0)	(55.0; 72.0)	(45.0; 66.0)
Anamnesis				
Family history of CAD, *n* (%)	49 (45.4)	44 (45.8)	31 (32.0)	0
Smoking, *n* (%)	58 (53.7)	78 (81.3)	80 (83.3)	0
Arterial hypertension, *n* (%)	97 (89.8)	84 (87.5)	64 (66.7)	0
Hypercholesterolemia, *n* (%)	15 (13.9)	12 (12.5)	16 (16.7)	0
History of MI, *n* (%)	33 (30.5)	39 (40.6)	8 (8.3)	0
Body mass index, kg/m^2^				
<25, *n* (%)	53 (49.1)	45 (46.9)	48 (50.0)	40 (100)
25.0–29.9, *n* (%)	45 (41.6)	42 (43.7)	40 (41.7)	0
30.0–39.9, *n* (%)	10 (9.3)	9 (9.4)	8 (8.3)	0
Coronary artery disease				
Atherosclerosis of the 1st coronary artery, *n* (%)	50 (46.3)	9 (9.4)	0	0
Atherosclerosis of the 2nd coronary artery, *n* (%)	36 (33.3)	21 (21.8)	0	0
Atherosclerosis of three or more coronary artery, *n* (%)	22 (20.4)	66 (68.8)	0	0
Ejection fraction, %				
>50, *n* (%)	76 (70.4)	90 (93.8)	88 (91.7)	40 (100)
40–49, *n* (%)	28 (25.9)	3 (3.1)	8 (8.3)	0
<40, *n* (%)	4 (3.7)	3 (3.1)	0	0
Treatment strategy/group of drugs				
Aspirin, *n* (%)	112 (98.2)	114 (95.0)	0	0
Clopidogrel, *n* (%)	114 (100)	18 (15.0)	0	0
Warfarin, *n* (%)	0	0	80 (83.3)	0
Heparin, *n* (%)	114 (100)	120 (100)	0	0
β-blockers, *n* (%)	114 (100)	108 (90)	86 (89.6)	0
ACE inhibitors, *n* (%)	102 (89.4)	90 (75)	74 (77.1)	0
Statins, *n* (%)	114 (100)	120 (100)	70 (72.9)	0
Calcium channel Blocker, *n* (%)	101 (88.6)	90 (75)	70 (72.9)	0
Nitrates, *n* (%)	20 (17.5)	6 (5.0)	10 (10.4)	0
Diuretics, *n* (%)	36 (31.6)	96 (80.0)	82 (85.4)	0

**Table 3 diagnostics-15-02772-t003:** Comparative characteristics of leptin and leptin receptor levels and FLI in serum of examined patients.

Variable	MI	Chronic CHD	AHD	Control Group	*p*
	1	2	3	4	
Leptin, ng/mL	11.31 [6.8; 22.6]	16.37 [7.0; 20.5]	3.54 [3.3; 9.1]	3.2 [2.7; 5.6]	*p* (1–2) = 0.4
					*p* (1–3) = 0.001
					*p* (1–4) = 0.001
					*p* (2–3) = 0.001
					*p* (2–4) = 0.001
					*p* (3–4) = 0.89
Leptin receptor, ng/mL	40.49 [29.3; 46.1]	34.82 [27.3; 47.8]	57.06 [41.6; 65.7]	58.06 [45.6; 67.7]	*p* (1–2) = 0.68
					*p* (1–3) = 0.003
					*p* (1–4) = 0.003
					*p* (2–3) = 0.001
					*p* (2–4) = 0.001
					*p* (3–4) = 0.91
FLI	31.91 [12.9; 65.5]	39.08 [19.1; 83.6]	6.04 [5.1; 22.4]	5.05 [4.2; 25.0]	*p* (1–2) = 0.3
					*p* (1–3) = 0.004
					*p* (1–4) = 0.003
					*p* (2–3) = 0.001
					*p* (2–4) = 0.001
					*p* (3–4) = 0.89

## Data Availability

Data is available from the authors upon reasonable request. The data are not publicly available due to privacy.

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
