# Peer review of "Systemic and Local Leptin Resistance in Patients with Cardiovascular Diseases"

_diagnostics, 2025, doi:10.3390/diagnostics15212772_

Round 1

Reviewer 1 Report

Comments and Suggestions for Authors

The cross-sectional study by Olga Gruzdeva and colleagues aimed to assess leptin resistance markers in serum and local cardiac adipose tissue depots, as well as the frequency of leptin resistance (LR) detection in patients with cardiovascular diseases (CVD). The study included 108 patients with myocardial infarction (MI), 96 patients with chronic coronary heart disease (CHD), 196 patients with acquired heart disease (AHD), and 40 healthy control participants. The manuscript presents interesting findings on LR in different types of cardiovascular diseases. However, several areas could be improved.

Majors:

  1. The statement in the first sentence, “The purpose of this study was to evaluate leptin resistance markers in serum and local cardiac adipose tissue depots as well as the rate of leptin resistance detection in patients with cardiovascular diseases (CVDs)” is somewhat vague and needs clarification. For example, it could be revised to say, "This study aimed to evaluate systemic and local markers of leptin resistance (LR) in patients with different forms of cardiovascular disease (CVD), and to determine the prevalence and tissue-specific expression patterns that contribute to LR”.
  2. The authors used the term "acquired heart disease (AHD)", which is broad and potentially confusing. Please clarify which specific conditions this group includes. Are they valvular diseases, cardiomyopathies, or others?
  3. The authors should clarify the rationale for the choice of investigating LR specifically in the cardiac adipose depots.
  4. Data for LR (leptin >6.45 ng/mL and FLI >25) are presented without justifications. Are these thresholds validated in previous literature or derived from your cohort?
  5. The abstract's results are presented without statistical data (p-values, SD, or confidence intervals), making it difficult to assess the strength of the findings. Authors should include changes with values indicating statistical significance.
  6. Were there any gender-specific differences in leptin or receptor expression?
  7. Were the observed LR patterns associated with any clinical outcomes (e.g., severity of CVD, complications)?

Minors:

  1. Too many abbreviations in the abstract. Consider limiting these to the most relevant ones.
  2. Rephrase for clarity: "MI and chronic CHD patients showed significantly elevated leptin levels and FLI, indicating a high prevalence of LR, unlike AHD patients”.
  3. For consistency, use either “chronic CHD” or just “CHD”.
Comments on the Quality of English Language

N/A

Reviewer 2 Report

Comments and Suggestions for Authors

1. Regarding the title: It is unclear whether the term "local" refers strictly to cardiac adipose tissue or also to other deposits.
2. Please provide a much better explanation of why you chose to include patients with MI, CHD, and AV stenosis in the study and why you did not just include patients with MI and CHD. You can add a diagram to highlight the pathophysiological processes involved.
3. Please add data on the sex of the patients included in the study to the descriptive table. Regarding the table, it would also be useful to add data about abdominal circumference, cholesterol levels, and the presence or absence of a diagnosis of metabolic syndrome.
4. Figure 3 is quite difficult to read, as the two graphs are quite crowded. Please make changes related to this aspect.
5. Some of the proposed mechanisms (SOCS3, PTP1B, GLUT4) are taken from animal model studies, but extrapolation to patients is not sufficiently supported.
6. You mentioned on line 366, "Chronic exposure to leptin promotes cardiomyocyte hypertrophy and fibrosis, exacerbating myocardial dysfunction"—please provide more data and studies to support this. Additionally, you have no citation at the end of this statement.
7. Please justify the leptin resistance thresholds by referencing international literature.
8. Line 276-284 - There is no citation for this text fragment, even though you are discussing data found in the literature. Please add the citation.

Reviewer 3 Report

Comments and Suggestions for Authors

The manuscript is devoted to an important topic of leptin resistance in patients with different cardia lesions. It presents valuable information that will be interesting to readers. However, certain aspects require improvement. 

Abstract

It is desirable to present some background information but not only the purpose of the study in the Background section.

Introduction

It is desirable to more clearly define what is known from the literature about leptin resistance in patients with cardiac lesions.

Material and Methods

It is not clear if this is prospective or retrospective study.

Clinical and demographic data (Table 1) should be presented in the Results section. It is desirable to present p values for all parameters included in Table 1.

Material and Methods section should more clearly define leptin resistance markers as well as how the leptin resistance frequency was determined. 

Discussion

This section is extremely lengthy and would benefit from shortening. Also, the number of references may be reduced.

If it is not a specific requirement of the journal, limitations of the study should be presented in the Discussion section but not separately.

Conclusions

According to the authors, "The purpose of this study was to evaluate leptin resistance markers in serum and local cardiac adipose tissue depots as well as the rate of leptin resistance detection in patients with cardiovascular diseases". Thus, conclusion should clearly reflect findings on leptin resistance markers and the frequency of leptin resistance in different cardiac lesions. Conclusion in the manuscript text is written in poor English. Conclusions in the Abstract and the manuscript text should match.    

Comments on the Quality of English Language

Review of the manuscript (especially the conclusion section) by the native language speaker would be recommended.  

Round 2

Reviewer 1 Report

Comments and Suggestions for Authors

No further comments.

Reviewer 2 Report

Comments and Suggestions for Authors

The article can be published. 

Reviewer 3 Report

Comments and Suggestions for Authors

No additional comments.